# Role of TLRs in HIV-1 Infection and Potential of TLR Agonists in HIV-1 Vaccine Development and Treatment Strategies

**DOI:** 10.3390/pathogens12010092

**Published:** 2023-01-05

**Authors:** Marija Rozman, Snjezana Zidovec-Lepej, Karlo Jambrosic, Maja Babić, Irena Drmić Hofman

**Affiliations:** 1Department of Immunological and Molecular Diagnostics, University Hospital for Infectious Diseases Zagreb, 10000 Zagreb, Croatia; 2Laboratory for Analytical Chemistry and Biogeochemistry of Organic Compounds, Division for Marine and Environmental Research, Ruder Boskovic Institute, 10000 Zagreb, Croatia; 3Srebrnjak Children’s Hospital, 10000 Zagreb, Croatia; 4School of Medicine, University of Split, 21000 Split, Croatia; 5Department of Health Studies, University of Split, 21000 Split, Croatia

**Keywords:** HIV-1, Toll-like receptors (TLRs), innate immunity, TLR agonists, latency reversing agents (LRAs)

## Abstract

Toll-like receptors (TLRs), as a family of pattern recognition receptors, play an important role in the recognition of HIV-1 molecular structures by various cells of the innate immune system, but also provide a functional association with subsequent mechanisms of adaptive immunity. TLR7 and TLR8 play a particularly important role in the innate immune response to RNA viruses due to their ability to recognise GU-rich single-stranded RNA molecules and subsequently activate intracellular signalling pathways resulting in expression of genes coding for various biological response modifiers (interferons, proinflammatory cytokines, chemokines). The aim of this review is to summarise the most recent knowledge on the role of TLRs in the innate immune response to HIV-1 and the role of TLR gene polymorphisms in the biology and in the clinical aspects of HIV infections. In addition, the role of TLR agonists as latency reversing agents in research to treat HIV infections and as immunomodulators in HIV vaccine research will be discussed.

## 1. Introduction

According to the Joint United Nations Program on HIV/AIDS (UNAIDS) data for 2021, there were 37.7 million [30.2 million–45.1 million] people living with HIV/AIDS at the end of 2020, the majority of whom live in eastern and southern Africa (an estimated 20.6 million, range 16.8–24.4 million) [1]. Despite a significant reduction in the number of new HIV-1 infections in the last decade, during 2020 a total of 1.5 million [1.0 million–2.0 million] persons were infected with HIV with 150,000 infections in children <15 years of age [1]. Despite effective antiretroviral therapies, infection with this virus led to an estimated 680,000 [480,000–1.0 million] deaths in 2020, emphasising the need for the continuous efforts in developing feasible strategies to treat these infections and develop an effective vaccine [1]. Although 40 years have passed since the isolation of HIV-1 by Barré-Sinoussi et al. (1983) and the significant achievements in basic and clinical science that followed, innate and specific immunity to HIV-1 remains an exceptionally important area of research [2]. Currently, HIV treatment is based on the combination of antiretroviral drugs from several classes designed to specifically target various stages of the virus replication cycle including reverse transcriptase inhibitors, protease inhibitors, integrase strand transfer inhibitors, capsid inhibitor, entry inhibitors (fusion inhibitor enfuvirtide and CCR5 inhibitor maraviroc), attachment inhibitors, and a CD4-directed post-attachment inhibitor monoclonal antibody [3]. Antiretroviral therapy (ART) is exceptionally successful in reducing the intensity of viral replication but fails to eradicate the virus and is unable to cure the infected individuals. To date, only some individuals have been cured of HIV-1 infection using stem cell transplantation from CCR5 delta 32 homozygotic donors [4,5,6]. Therefore, even in successfully-treated patients, HIV establishes latent reservoirs, most frequently in memory CD4+ T cells, where it stays in its dormant state and is not visible to the immune system. The presence of a latent viral reservoir is the key obstacle to curing HIV. Several therapeutic strategies (including the “shock and kill” approach) designed to activate HIV-1 replication in latently infected cells and allow their recognition and elimination by the immune system, have been evaluated [7]. Eradication of latent reservoirs includes the use of latency reversing agents (LRA) including protein kinase C, histone deacetylase inhibitors, and toll-like receptor (TLR) agonists [8]. TLRs play an important role in both innate and adaptive immunity and are responsible for the recognition of structurally conserved molecular patterns from various pathogens. Activation of TLRs initiates several signalling cascades resulting in the synthesis of proinflammatory cytokines and interferons [9]. The aim of this review is to summarise the current knowledge on the role of TLRs in the recognition of HIV-1 molecular patterns and activation of innate immune responses to the virus. In addition, the role of TLR agonists in current and future clinical trials leading to HIV-1 vaccine and development of novel therapeutic strategies leading to a cure will be evaluated.

## 2. HIV

HIV-1 is classified as viral species within the family *Retroviridae*, subfamily *Orthoretroviriane* and genus *Lentivirus* that also includes HIV-2 and simian immunodeficiency virus (SIV) that infects non-human primates and is an essential experimental model for HIV immunology research in vivo [10]. HIV-1 viral particles are approximately spherical with variable diameters of ∼145 nm consisting of a lipid bilayer envelope with 72 trimers of viral env glycoprotein (gp) 120 (gp120) attached to its surface by the viral transmembrane protein gp41; gp41 covers an internal hexameric lattice consisting of p17 viral matrix protein that interacts directly with the lipid envelope [11] (Figure 1). The fullerene cone-shaped capsid consists of p24 capsid protein (CA) that contains two single-stranded (+) genomic RNA molecules, two tRNA primers, viral enzymes (reverse transcriptase, protease, integrase), and regulatory proteins (Figure 1) (for a detailed review see Troyano-Hernáez et al., 2022, [12]).

The HIV genome is 9.75 kb in size and includes 9 open reading frames (ORF) which code for 15 proteins in total [13]. The *gag* ORF encodes a Pr55Gag precursor for the viral structural proteins: p24 capsid protein, p17 matrix protein, p7 nucleoprotein, and p6 protein that is required for the release of new viral particles. The *pol* ORF encodes the Pr160GagPol precursor for the viral enzymes: p10 protease, p51 reverse transcriptase, p5 RNase H, and p32 integrase. The *Env* ORF encodes the envelope precursor protein PrGp160 that is processed by cellular proteases into gp120 that binds the virion to the receptor and co-receptors and a transmembrane gp41 that enables viral fusion with the target cell. The HIV-1 genome also contains genes coding for regulatory proteins including tat, rev, nef, vif, vpr, vpu, and tev. The genome also contains long terminal repeats in the R3-U-R5 orientation [11,12]. However, genomes are mostly provided for HIV-1 subtype C distributed in India [14].

HIV-1 replication begins with the interaction between gp120 and the viral receptor CD4 on the surface of target cells (most frequently activated memory CD4+ T-cell), followed by an interaction with a co-receptor (chemokine receptors CCR5 or CXCR4). Following the gp41-mediated fusion of the viral and cellular membranes, the viral capsid traffics within the cytosol where it is imported into the nucleus (Figure 2) [15]. Reverse transcription that converts viral RNA into linear double-stranded DNA is completed in the nucleus and is followed by capsid uncoating and proviral DNA integration into the cell’s genome. The RNA genome is reverse transcribed into dsDNA by the viral reverse transcriptase that generates both strands of DNA using the genomic RNA template for first strand synthesis and the resulting cDNA for second strand synthesis, while the (integrated) dsDNA genome is transcribed into viral RNA by the host RNA polymerase II. Activation of transcription from the integrated proviral DNA enables the expression of GagPol and Gag polyproteins that assemble into hexameric membrane-associated lattices. Dimers of genomic viral RNA and gp120 trimers are incorporated into newly-formed immature particles that begin to mature as the viral protease cleaves the GagPol and Gag polyproteins into functional viral proteins [16]. Upon formation of p24 hexamers and pentamers, a viral capsid is formed and subsequently, mature virions are formed.

## 3. Toll-Like Receptors 

Prior to the discovery of the TLRs, the mechanism of innate immunity was inconclusive and initially considered as only a tool for the activation of the adaptive immune response. However, the discovery of TLRs identified a mechanism for cytokine and interferon induction, and, consequently, was awarded a Nobel prize in Medicine in 2011 [17]. TLRs are pattern recognition receptors (PRR) that also includes nucleotide oligomerization domain-like receptors (NOD-like receptors), retinoic acid-inducible gene-like receptors (RIG-like receptors), and other molecules [18]. TLRs are categorised as receptors of the innate immune response as they are part of the first line of immunological defence, but their activation has an impact on both innate and adaptive immunity. They recognise structurally conserved molecules known as pathogen associated molecular patterns (PAMPs) such as proteins, peptides, nucleic acids, or phospholipids from various organisms which represent the exogenous ligands for TLRs [9]. In addition, TLRs can be activated by endogenous ligands characterised as damage associated molecular patterns (DAMP) such as extracellular matrix components, heat shock proteins, and parts of cells from apoptotic or damaged cells [19]. In addition, some TLRs are activated by nucleic acid ligands such as single-stranded RNA (ssRNA), double-stranded RNA (dsRNA), and DNA with unmethylated CpG motifs in order to sustain host cell protection against viruses [20].

### 3.1. Types of TLRs in Humans

To date, there are 13 known TLRs among which 10 are functional in humans and 12 in mice [21]. In humans, TLR1, TLR2, TLR4, TLR5, TLR6, and TLR10 are expressed on the cell surface while TLR3, TLR7, TLR8, and TLR9 are present on the endosomal membrane [22]. *TLR11* is present the human genome as well but its expression is restricted due to the presence of stop codons [23]. However, in mice, TLR11 forms a heterodimer with TLR12 during recognition of the protein profilin from *Toxoplasma gondii* which leads to the activation of Myd88-dependent signalling pathways [24]. Heterodimers TLR1/TLR2 and TLR2/TLR6 recognise triacylated and diacylated lipopeptides from bacteria [25]. TLR2 as a homodimer recognises a wide range of PAMPs such as lipopolysaccharides, peptidoglycan, lipoproteins, and lipoteichoic acid [26]. TLR4 is activated by bacterial lipopolysaccharides as well as the synthetic compound monophosphoryl A (MPLA) [27]. TLR5 is the only TLR in humans that is activated by bacterial flagellin, which is its only ligand [28] (Figure 3). TLR3, TLR7, TLR8, and TLR9 are activated by viral RNA and DNA [29]. Among the TLRs in humans, TLR10 was previously considered an orphan receptor since its ligands could not be identified. However, recent studies have demonstrated its higher expression in primary human breast milk cells infected with HIV-1 suggesting that certain HIV-1 structural and envelope proteins act as ligands for TLR10 [30]. On the other hand, it has been shown that blocking of TLR10 with specific antibodies increases TLR2 function and, consequently, cytokine synthesis, which indicates its role as an inhibitory receptor [31]. 

### 3.2. Structure and Position

TLRs are expressed on dendritic cells, macrophages, NK cells, B-cells and T-cells, epithelial cells, endothelial cells, and fibroblasts [18]. TLRs are present on the cell surface and on the endosomal membrane. They are classified as type I transmembrane receptors with a ligand-binding extracellular N-terminal domain, transmembrane region, and cytosolic signalling C-terminal domain (Figure 2). The extracellular region of TLRs contains leucine-rich repeat (LRR) motifs which recognise PAMPs and DAMPs [32]. The LRR motif is found in many proteins that participate in immune recognition in plants and animals. The LRR region itself contains an N-terminal domain and C-terminal domain which are positioned outside the cell [33]. Additionally, TLR7–9 contain a Z-loop positioned in the region between LRR14 and LRR15 where proteolytic cleavage occurs, which is necessary for their activation [34]. Proteolytic cleavage of TLR7–9 is a prerequisite for the formation of dimers which are necessary for the recruitment of TIR-domain adapter proteins [35]. The intracellular signalling domain is known as the Toll IL-1 Receptor domain (TIR domain) which, upon activation, recruits adaptor molecules and initiates signalling cascades such as the nuclear factor kappa B (NF-kB) pathway [18]. According to their sequence homology, TLRs can be grouped into several subfamilies: TLR3, TLR4, TLR5, TLR7/8/9, TLR1/2/6/10, and TLR11/12/21/21/22/23 [36].

### 3.3. Mechanism of Action

TLR activation starts with the binding of a ligand which causes dimerization of two receptors. The majority of TLRs form homodimers, with exception of TLR1 and TLR6 that form heterodimers with TLR2. In addition, some TLRs need additional coreceptors for their complete functionality. For example, TLR4 requires myeloid differentiation factor 2 (MD-2) to efficiently recognise lipopolysaccharides on Gram-negative bacteria [37]. Receptor dimerization recruits TIR domain-containing adaptor molecules such as myeloid differentiation primary response protein 88 (MyD88), MyD88 adaptor-like (MAL), TIR domain-containing adapter-inducing interferon-β (TRIF) and TRIF-related adaptor molecule (TRAM) to the cytoplasmic TIR domain of TLRs. The signalling pathways initiated by adaptor molecules can be MyD88-dependent or MyD88-independent [18]. In the MyD88-dependent pathway, MyD88 interacts with interleukin receptor-associated kinase 4 (IRAK4) using their N-terminal death domain. This interaction forms a complex called the myddosome which recruits other members of IRAK family [38]. The MyD88-dependent signalling pathway activates NF-κB and mitogen-activated protein kinases (MAPKs) which causes generation of proinflammatory cytokines [22]. On the other hand, TLRs which do not use the MyD88-dependent pathway use the TRIF-signalling pathway. In this pathway, the TRIF N-terminal domain recruits inhibitor of nuclear factor kappa-B kinase ε (IKKε) and TANK-binding kinase 1 (TBK1) which leads to the phosphorylation of IRK3 and, consequently, expression of proinflammatory cytokines and type I interferons (IFNs) [39]. TLR3 is the only receptor that uses only the TRIF-dependent signalling pathway. TLR4 is able to induce both pathways while other TLRs activate only the MyD88-dependent signalling pathways. A cascade of activated signalling pathways results in the activation of transcription factors such as interferon regulatory factors which leads to the synthesis of cytokines essential for the immune response to pathogens [33] (Figure 3). 

### 3.4. Basic Role of TLRs in Host Immunity

TLRs are present on a wide range of cells including immune and non-immune cells, but mostly importantly on dendritic cells (DC) and macrophages where they are expressed at high levels with noticeable variations among different cell subsets [40]. DCs express all the types of functional TLRs in humans, with TLR10 being present only on plasmacytoid DCs. The signalling pathways activated by TLRs are necessary for direct DC activation and, consequently, their maturation inside the lymph nodes [41]. Besides their activation through PAMPs, DCs can mature indirectly via inflammatory mediators released from other cells. However, negative-strand RNA virus-induced DC maturation is not dependent on TLRs. Research on mice myeloid DCs shows their maturation in the presence of virus infection but, at the same time, in the absence of NA-sensing TLRs [42]. Fewer TLRs are expressed on T cells with variable results in different species. In human CD4+ T cells, mRNA expression for TLR1, TLR2, TLR3, TLR4, TLR5, TLR7, and TLR9 was detected, but no protein expression was noted. On the other hand, expression of the mRNA for TLR2, TLR3, TLR4, and TLR5 was detected in CD8+ T lymphocytes with functional proteins being expressed as well [40]. On activated T-cells, TLR2 is expressed as a costimulatory receptor that is responsible for the maintenance of memory T-cells [43]. Expression of TLR2 on human T cells is amplified by the activation of the T-cell receptor (TCR) which is necessary for the recognition of an antigenic peptide presented on the major histocompatibility complex (MHC). This association probably represents a mechanism that sustains the need for the stronger immune reaction in the presence of pathogens, since TLR2 works as both homodimer and heterodimer with a wide range of ligands. Naive B cells express low levels of all TLRs while their levels are highly expressed on activated B cells and memory cells. The expression of TLR9 and TLR10 is additionally upregulated after stimulation of the B-cell receptor (BCR) or costimulatory molecule CD40 [44]. In addition, TLR signalling is important for the development of select cellular subpopulations during haematopoiesis. For example, TLR activation in haematopoietic stem cells (HSC) directs their development towards myeloid cells rather than lymphocytes. In common lymphoid progenitor cells (CLPs), TLR signalling causes differentiation into dendritic cells rather than B cells. Furthermore, once transitional B cells emerge, signalling by TLRs is necessary for their differentiation into mature B cells, plasma cells, or memory cells [45]. 

#### Recognition of Nucleic Acids by TLRs

TLRs can provide protection against viruses as well. Contrary to bacteria, viruses usually do not have recognizable molecular patterns on their surface. However, haemagglutinin from influenza virus H5N1 is a ligand for TLR2, TLR3, and TLR4 [46] and envelope proteins from respiratory syncytial virus (RSV) and mouse mammary tumour virus (MMTV) are ligands for TLR4 [47,48]. In addition, HIV-1 structural proteins p17, p24, and gp41 act as ligands for TLR2. The majority of nucleic acid (NA)-sensing TLRs recognise viral genomes. All NA-sensing TLRs contain the double-pass transmembrane protein Unc-93 homologue B1 (UNC93B1) which guides them to the endosomes after their synthesis on the endoplasmic reticulum [49]. UNC93B1 facilitates TLR stability, cleavage, and ligand recognition, and guides TLR5 to the cell surface [50]. In addition, UNC93B1 moves away from TLR9 and TLR3 upon ligand binding in order for them to dimerise and activate signalling. Each TLR has a precise threshold which avoids sensing of self-nucleic acids and, consequently, decreases chances for autoimmune disease development. On the other hand, the threshold must not be too high so that it would not activate the immune response when needed [51]. TLR3 recognises double-stranded RNA (dsRNA) and, contrary to other NA-sensing TLRs, is present both on the cell surface and on the endosomal membrane of macrophages and mast cells [52]. TLR7 and TLR8 recognise ssRNA and its breakdown products, nucleosides and oligonucleotides, while TLR9 recognises ssDNA with unmethylated CpG motifs [20]. In order for TLR7-9 to recognise NAs, they first have to be processed by RNases and DNases. On the contrary, RNases negatively regulate the TLR3 response since its ligand is a dsRNA. Therefore, NA processing inside the endosomal compartment acts both as positive and as negative regulators of TLR activation. Endosomes have a pH of approximately 5.0 which is optimal for the function of DNase II and RNase T2; therefore, these enzymes are the ones responsible for DNA and RNA degradation inside endosomes [53].

### 3.5. The Role of TLR Receptor Polymorphisms in the Biology and in the Clinical Aspects of the HIV Infection

The TLR genes contain several single-nucleotide polymorphisms (SNPs) that have been linked to altered susceptibility to inflammatory and infectious diseases, including HIV infection. Complex associations between genetic variants of genes coding for TLRs and environmental factors have been revealed by cutting-edge genomic techniques and important advancements in our understanding of innate immune functions. However, interpretations of how TLR SNPs affect the immune response are further complicated by other factors including ethnicity, gender, and the possibility of multigenic effects. The molecular mechanisms by which SNPs influence TLR receptor functioning are unclear for many TLRs [54]. Recent research on TLR polymorphisms and their correlation with various outcomes, published over the last five years, is presented in this review (Appendix A).

#### 3.5.1. TLR1 (*4p14*) and TLR2 (*4q31.3*)

The exact mechanism by which the TLR1 polymorphism affects the HIV status is not clearly established. In has been shown that SNPs in *TLR1* (rs5743551 and rs5743618), as well as in *TLR4*, *TLR6*, and *TLR8*, are associated with the HIV status, and these associations appear to be race-specific [55]. 

Proximal promoter deletion (−196 to −174 or rs111200466) in the *TLR2* gene has been the focus of several studies. According to a study of an Indian cohort, Vidyant et al. have shown that the non-synonymous polymorphisms Arg753Gln, Arg677Trp, and Pro631His were uncommon, but the *TLR2* −196 to −174Ins/Del polymorphism was found to be a risk factor for HIV-1 infection (a mutant genotype and allele *TLR2* Del was more frequent in HIV-infected patients) [56]. However, a subsequent study by Royo et al. (2018) discovered a Hardy–Weinberg discordance in the data of Vidyant et al. and showed that a deletion in *TLR2*, rs111200466, protects against HIV-1 infection [57]. Their study in a Spanish cohort also showed that the deletion *TLR2* allele (196 to 174 Ins/Del) is linked to a higher likelihood of HIV infection and more rapid disease progression. However, some studies emphasise that a role of *TLR2* in HIV-1 disease progression rates (CD4+ 200 cells/μL) may be distinct from this gene’s impact on infection susceptibility [58]. In 2020, Shi et al. conducted a systematic review and meta-analysis of 10 studies covering 20 genes and showed a correlation between clinical findings in HIV-infected persons and *TLR2* −196 to −174 Ins/Del [54]. In 2022, *TLR2*, *TLR4*, and *TLR9* genetic diversity in HIV-1-infected patients with and without tuberculosis coinfection was studied by Kaushik et al. However, no significant differences in the frequency of the C2180T allele between the healthy and patient subcategories were found. Likewise, comparing *TLR2* alleles (2180 C/T) across various patient groups failed to find statistically significant changes in their frequency [59]. *TLR1* rs5743563 T/T, *TLR2* rs3804099 T/T, *TLR6* rs3796508 G/A, and *TLR6* rs3804099 C/T were associated with a higher risk of cryptococcal meningitis in HIV-negative patients, whereas rs5743563 and rs3804099 were associated with CSF cytokine expression [60]. 

#### 3.5.2. TLR3 (*4q35.1*)

In a study by Huik et al., the *TLR3* SNP rs3775291 T allele showed a protective effect against HIV infection among Caucasian intravenous drug users [61]. 

#### 3.5.3. TLR4 (*9q33.1*)

The majority of studies have focused on two cosegregating SNPs—Asp299Gly and Thr399Ile—within the *TLR4* gene. These SNPs are present in approximately 10% of Caucasians, and have been found to be more frequent in patients with infectious diseases, but it is unclear whether TLR4 can act as a direct sensor of retroviral particles. The *TLR4* Asp299Gly heterozygous genotype and the mutant allele G were discovered to be substantially more prevalent in HIV-1 infection than in healthy controls and to be related to stage progression by Vidyant et al. in 2019 [62]. 

The Thr399Ile polymorphism was not associated with HIV infection or stage progression. The authors also noted that patients with HIV infection were considerably more likely to carry the mutant G allele in *TLR4* (rs4986790), while an association with HIV infection was not shown for *TLR4* (rs4986791) [62]. A meta-analysis on the association between *TLR4* polymorphisms and HIV-1 was carried out by Kim YC et al. in 2020, as well as an association analysis using a matched control group collected from the 1000 Genomes Project. A significant correlation between the *TLR4* rs4986791 SNP and HIV infection risk was found in a matched control population and in HIV-infected patients from earlier studies. The *TLR4* rs4986791 G allele was also discovered to be a significant risk factor for HIV infection among Caucasians and four other populations examined in this meta-analysis [63]. In a systematic review and meta-analysis by Shi et al. in 2020 and Kim and Jeong in 2020, found that *TLR4* rs4986790 was associated with HIV infection [54,64]. TLR gene variants have been linked to increased or decreased susceptibility to a variety of infectious illnesses in several studies. According to a recent study by Kaushik et al. (2022), the *TLR4* Asp299Gly gene polymorphism is associated with increased susceptibility to active tuberculosis in HIV-infected patients from a north Indian population sample [59]. However, Mungmunpuntipantip et al. (2022) suggested that a number of problematic factors should be taken into account [65]. Aside from the Asp299Gly polymorphism in the *TLR4* gene, there are additional genetic variations that could be related to HIV and TB co-infection (e.g., rs12722 COL5A1 and rs3751143 P2X7) [65]. The *TLR4* 896A/G and *TLR9* 1174G/A polymorphisms are associated with the risk of infectious mononucleosis [66]. It is interesting to note that the *TLR4* Asp299Gly polymorphism has been linked to the development of cardiovascular illnesses in HIV-positive individuals [67].

#### 3.5.4. TLR6 (*4p14*) and TLR7 (Xp22.2)

Increased HIV expression, a positive correlation between plasma viral load, and the fact that TLR6 SNPs have not yet been linked to HIV infection and/or disease progression in North Americans suggest that these correlations may be race-specific [55]. The fact that TLR7 binds to retroviral ssRNA has drawn a lot of attention to its function. Increased viral loads and altered CD4+ T cell numbers are linked to a polymorphism in the *TLR7* [68,69]. Zhang et al. examined the impact of three *TLR7* intronic polymorphisms on HIV-1 infection susceptibility and progression in the Chinese MSM (men who have sex with men) population. MSM patients were shown to have considerably lower frequencies of the *TLR7* rs179010 allele T. Lower susceptibility to acute HIV-1 infection was linked to the haplotype TTA (patients with acute infections had slower disease progression and lower viral loads). The TLR7 rs179009 allele A was related to rapid progression, while rs179009 minor allele G was related to chronic infection [70]. Shaikh N. et al. found that TLR7 rs2074109 may be linked to HIV infection, and the G allele was substantially more prevalent in healthy controls. The same SNP may also have a negligible impact on HIV susceptibility and may be linked to a reduced probability of transmission [71]. The *TLR7* rs179008 T allele risk effects for HIV infection have been linked to high viral loads, lower CD4+ T-cell counts, and a quicker development to advanced immunological suppression in Caucasian HIV patients [72]. There was no such correlation in the Indian cohort [68]. In a systematic review by Shi et al., *TLR7* rs179008 and rs2074109 were linked to HIV infection. The T allele of *TLR7* at rs179008 increased the likelihood of HIV infection [69]. *TLR7* polymorphisms in treatment-naïve HIV-infected persons were investigated by Singh et al. in 2020. There were no discernible differences in the genotype and haplotype of the *TLR7* Gln11Leu (A/T) and IVS2-151 (A/G) polymorphisms between HIV-infected persons and healthy controls. The *TLR7* rs179009AG and AG genotypes were found to be less common in HIV-infected people. Their higher prevalence in healthy persons suggests an association with a lower risk for HIV-1 infection [73].

#### 3.5.5. TLR8 (*Xp22.2*) and TLR9 (*3p21.2*)

*TLR8* rs3764880 has been shown to have a protective effect against progression of HIV disease [68]. Beima-Sofie et al. showed a correlation between the peak plasma HIV-1 RNA levels and *TLR8* 1A/G and the haplotype-tagging *TLR7* variation (rs1634319) [74]. Valverde-Villegas et al. found that the polymorphisms in the *TLR7–9* genes contributed to the HIV infection susceptibility in Brazilians with European and African ancestry, highlighting the impact of ethnic background on HIV infection susceptibility [75]. Numerous infectious diseases, including HIV infections, have been associated with polymorphisms in the *TLR9* gene and promoter area, including *TLR9* 1635A/G and 1486C/T [68,69]. Joshi et al. showed that *TLR9* rs352140 was related to lower CD4 T cell counts, higher viral loads, and faster disease progression in HIV-infected persons. Lower CD4+ T cell counts, higher CD8+ T cell activation, and higher IP-10 levels were all weakly correlated with the *TLR9* 1486C/T polymorphism. These findings offer a new perspective on HIV-mediated immune activation and CD4+ T cell depletion since it is well established that TLR9 stimulation induces IP-10 synthesis in dendritic cells. The *TLR9* SNPs 1635A/G and 1486C/T may be linked to disease progression, and TLR9 stimulation by viral CpG DNA is an important step in the immunopathogenesis of HIV disease [76]. As recently noted by Vallejo et al., the association between the *TLR9* 1635AA genotype is associated with increased probability of HIV-1 rebound after ART interruption [77]. *TLR9* polymorphisms are associated with the clinical presentation of HIV disease and the GG genotype has been linked to an increased risk of HIV infection [54,78,79]. A study by Shaikh et al. found that the *TLR9* rs352140 variant is not associated with HIV disease progression [71]. However, significant differences in the allele frequency were found between the HIV+/HCV+ group and controls by Kulmann-Leal et al., with the GG genotype acting as a protective factor [79]. Valverde-Villegas et al. showed that *TLR9* rs5743836 C carriers of European descent had an increased susceptibility to HIV infection, but the CT genotype in people of African descent was linked to protection from HIV infection. Additionally, African descendants’ vulnerability to HIV+/HCV+ co-infection was linked to the *TLR9* rs352140 AA variant genotype [80]. According to Jablonska et al., *TLR9* polymorphisms are linked to higher cytomegalovirus and Epstein-Barr virus uptake in HIV-exposed infants [81]. An increased prevalence of the A allele was also seen in ART-naive HIV+ patients who developed active TB, according to Kaushik et al. [59]. Varshney D et al. showed that TLR gene polymorphisms may be linked to TB susceptibility and showed that polymorphisms in the *TLR1–2*, *TLR4*, *TLR6* as well as *TLR9* genes have a protective effect in particular ethnic populations [82].

#### 3.5.6. Significance of TLR Polymorphisms

Based on the information provided in this review, it can be concluded that more research is necessary to clarify the part that TLR polymorphisms play in the onset, clinical course, and treatment of HIV-1 infections, as well as in vaccine design. To further explore this topic, larger studies are required, which should involve varied demographic and clinical characteristics for populations of various ancestries.

## 4. Activation of Latent HIV-1 Reservoir

HIV-1 can initiate latency in different types of cells including naive CD4+ T cells, stem memory T lymphocytes, macrophages, myeloid cells and haematopoietic stem cells. However, the largest HIV-1 latent reservoir is inside central memory CD4+ T cells. Memory cells within the latent HIV-1 reservoir have a compact chromatin structure, maintained by de-acetylation and methylation of histones. The lack of transcription provides an optimal environment for HIV-1 latency [83]. When HIV-1 DNA incorporates into the genome as a provirus, it can disrupt cell signalling pathways and, consequently, cause comorbidities. During latency, HIV-1 transcription factors are either lacking or present in their inactive form and the lack of active transcription allows HIV to escape immune system recognition during ART [22,84]. In order to activate the HIV-1 reservoir, several latency reversing agents (LRAs) have been evaluated including histone deacetylase inhibitors (HDACi), protein kinase C (PKC) agonists, and benzotriazoles [85]. The use of LRAs is part of “shock and kill” therapy which consists of two phases: activation of HIV-1 transcription by an LRA to make viral proteins exposed (“shock”) followed by their elimination of the immune system (“kill”). All LRAs work in a specific way to induce transcription in initially transcriptionally silent cells. Histone deacetylases remove the acetyl groups from acetylated lysines on histones, and this modification makes chromatin more compact and downregulates gene expression. Therefore, their inhibitors, such as valproic acid, vorinostat, and givinostat, are used as LRAs to activate the latent HIV-1 reservoir [86]. HDACi represent a promising HIV-1 therapy candidates since their use induces an increase in the expression of HIV-1 mRNA, notably without the elimination of latently-infected cells [87]. PKC agonists, such as ingenol-3-angelate, bryostatin-1, or prostratin, activate transcription factors, namely NF-κB, which binds to HIV-1 long terminal repeats (LTRs) and initiates mRNA transcription [88]. PKC agonists were successful in reactivating HIV-1 latency. However, during clinical trials of these compounds as anticancer drugs, important side effects such as fever and headaches have been observed [83]. Benzotriazoles have shown promising results in activating HIV-1 both in vitro and ex vivo and managed to induce HIV-1 reactivation without further cell proliferation. Benzotriazoles block attachment of proteins from the small ubiquitin-like modifier (SUMO) family in the process called SUMOylation, on signal transducer and activator of transcription 5 (STAT5) proteins [89]. There are two isoforms of STAT5, STAT5a and STAT5b, that both regulate the transcription of genes which is initiated by cytokines or tyrosine kinase receptors. STAT5 proteins are activated by tyrosine phosphorylation and deactivated by SUMOylation. Therefore, when SUMO family proteins are blocked, STAT5 proteins cannot deactivate, leaving these proteins in their active state which induces HIV-1 reactivation [90]. However, benzotriazoles were only studied using cultured latently infected peripheral blood mononuclear cells (PBMCs) as a model and should be further evaluated in human clinical trials [89].

### TLR Agonists as Latency Reversing Agents

TLR agonists can be used in order to reactivate latent HIV-1 reservoirs and make it visible to the immune system [22]. Investigations on the role of TLR agonists in HIV-1 infection have focused on two main areas of interest: their role as LRAs in novel treatment strategies to functionally cure HIV infections and their potential as immune system adjuvants in HIV vaccine research. The majority of studies that provided a scientific foundation for the potential usefulness of TLR agonists as LRAs evaluated the ability of these compounds to reactivate latently infected cells harbouring replication-competent HIV-1 by measuring viremia in the supernatant of PMBCs isolated from successfully treated HIV-infected patients with undetectable plasma viremia. In HIV-vaccine research, the ability of TLR agonists to enhance HIV-specific cellular immunity, induce immunophenotypic changes in HIV-specific CD4+ and CD8+ T cells, induce intracellular cytokine synthesis and antibody-mediated clearance of HIV-infected cells, and the ability to activate and induce cytokine synthesis in NK cells and DCs have been extensively evaluated. The majority of research on HIV-1 reactivation has been performed in models analysing NA-sensing TLRs with a particular emphasis on TLR7 [91]. Humanised mouse models have been used to define a relationship between TLRs and latent HIV-1 reservoirs. Mouse models of HIV infection and animal studies of SIV infection both represent excellent research tools for this area of research.

Studies have shown that activation of PAMPs can transactivate the promoter of HIV-1 LTRs [92]. The first TLR agonist which was shown to efficiently inhibit HIV-1 replication ex vivo was Gene Expression Modulator 91 (GEM91). GEM91 is an antisense oligonucleotide phosphorothioate complementary to the mRNA for the structural Gag protein in HIV-1 [93]. However, this agonist showed contradictory results, displaying both reduction of HIV-1 replication in peripheral blood cells in humans but, simultaneously, an increase in viremia [22]. The increase in viremia was associated with the stimulation of TLR9 due to the high amount of CpG motifs in GEM91. However, reduction in HIV-1 replication was achieved which is in accordance with the hypothesis that GEM91 can be used for the termination of translation of Gag protein and, consequently, lead to the reduction of synthesised virions. To date, many TLR agonists were tested in order to reactivate latent infection. A novel potent molecule SMU-Z1 was identified as a TLR1/2 agonist with efficient enhancement of HIV-1 transcription in vitro and ex vivo in PBMCs from aviremic HIV-1 patients receiving ART [94] Additionally, SMU-Z1 increased the immune response and expression of TLR2 in PBMCs, decreased HIV-1 replication, and promoted degranulation and interferon gamma synthesis (IFN-γ) in NK cells [94]. Besides SMU-Z1, many synthetic ligands for TLRs that were synthesised based on the structure of TLRs natural ligands, have been used and tested as agonists. For instance, synthetic lipopeptides have been used as a TLR2 agonist, Pam2CSK4 is a TLR2 homodimer and TLR2/TLR6 agonist which imitates diacylated lipopeptide while Pam3CSK4 is a TLR1/TLR2 agonist, which is synthesised as triacylated lipopeptide [92]. A TLR7 agonist (GS-9620) increased extracellular HIV-1 RNA in PBMCs of HIV-1 patients on ART and increased CD4+ and CD8+ T cell activation [95]. For additional confirmation, GS-9620, used as TLR-7 oral agonist in chimpanzees, also suppressed hepatitis B virus [96]. The TLR7/8 agonist R-848 showed promising results in HIV-1 reactivation from cells of myeloid/monocytic origin. Further research on this agonist defined two ways that signalling pathways activated by TLR8 lead to HIV-1 reactivation. TLR8 activates HIV-1 via TNF-α in CD4+ T cells in an autocrine and paracrine manner, and via the MAPK pathway in cells of myeloid/monocytic origin [97]. The TLR9 agonist tMNG1703, synthesised from natural DNA to imitate the CG motifs in TLR9 natural ligands, has been used in PBMCs from HIV-1 infected patients. The results showed increased activation and function of NK cells and HIV-1 transcription in CD4+ T cells. Additionally, MNG1703 is currently in phase III clinical trial as a therapy for metastatic colorectal cancer. Previous phase trials results showed success in both latency reversal and immunity induction [98].

#### Studies in Non-Human Primates and Mice

In vivo studies in non-human primates (NHP) and mice evaluated the role of TLR3, TLR7, and TLR9 agonists as potential LRAs in clinical research focused on curing HIV-1 (Table 1). Cheng et al. (2018) evaluated the TLR3 agonist poly(I:C) that has previously shown promising results ex vivo, in combination with a CD40-targeting HIV vaccine in humanised mice [99]. The concept of the vaccine was based on targeting CD40 that induces maturation of dendritic cells and presentation of the vaccine antigens to specific immune mechanisms. The TLR3 agonist poly (I:C) activates myeloid DCs in vitro and in vivo and increases CD40 expression on human antigen presenting cells. Co-administration of the αCD40.HIV5pep vaccine and TLR3 agonist induced HIV-specific T-cell responses, reduced the size of latent HIV reservoirs, and enabled a better control of virus replication after the cessation of antiretroviral therapy [99]. Despite the evidence on the ability of the TLR3 agonist to reactivate HIV-1 reservoirs in humanised mice, this class of agonists was not further evaluated in NHP.

The ability of TLR7 agonists to act as LRAs in NHP have been initially investigated by Lim et al. (2018) and Del Prete (2019). Lim et al. (2018) demonstrated the ability of TLR7 agonists to induce viral replication in treated animals (up to 1000 SIV RNA copies/mL) and to reduce the proviral SIV DNA reservoir by an average of 75% but failed to demonstrate a difference in time to viral rebound following cessation of antiretroviral therapy [100] (Table 1). Contrary to these findings, Del Prete et al. (2019) failed to demonstrate the effect of a TLR7 agonist on SIV replication or on the size of the proviral reservoir [101]. These results were most likely associated with important differences in the experimental concepts of the two studies. The potential role of TLR7 agonists as LRAs in novel strategies leading to a functional cure has been evaluated in studies conducted on non-human primates using env-specific, broadly neutralizing antibodies (bNAb) [22]. The hallmark study in this field was published by Borducchi et al. (2018) which showed that a combination of a TLR7 agonist and the V3-glycan-dependent bNAb PGT121 significantly delays viral rebound after antiretroviral treatment discontinuation in NHP infected with SIV-SF162P3 [102] (Table 1). The evaluation of residual replication-competent virus in adoptive transfer studies using PMBCs and lymph node mononuclear cells from five animals that did not experience a rebound after treatment discontinuation showed that SIV infection was not transferred to an uninfected host. In addition, the viral load in these animals remained undetectable upon depletion of CD8+ T cells [102]. This study clearly demonstrated the ability of TLR7 agonists to act as LRAs in vivo in combination with bNAb to successfully target latent viral reservoirs. More recently, Hsu et al. (2021) showed that a combination of a TLR7 agonist (GS-986) and two bNAbs (N6-LS and PGT121) induced Simian-Human Immunodeficiency virus (SHIV)-specific T-cell immunity and immune activation in *Macaca mulatta* receiving antiretroviral therapy starting at day 14 after inoculation with SHIV-1157ipd3N4 [75] (Table 2). Although only a modest effect on viral rebound was observed in this study (and no animals maintained control of viral replication), an effect of TLR7 on the augmentation of HIV-specific immunity was confirmed in vivo. Following these promising results, Mold et al. (2022) evaluated the effect of a TLR7 agonist and bNAb PGT121 in a different experimental setting regarding the initiation of treatment. Combination of a TLR7 agonist and bNAb was able to prevent viral rebound after treatment discontinuation in a subset of *Macaca mulatta* infected with SHIV-SF162P3 and treated since the chronic stage of infection [103] (Table 1). A promising role of TLR7 agonists as adjuvants in HIV vaccine research has been shown by another hallmark study by Borducchi et al. (2016) in Nature [104]. Therapeutic vaccination of *Macaca mulata* with SIVmac251 with a recombinant adenovirus serotype 26 (Ad26) vector expressing SIVsmE546 Gag/Pol/Env as the prime vaccination and a modified vaccinia Ankara (MVA) boost in combination with the TLR7 agonist GS-986 showed delays in time to viral rebound following treatment discontinuation, as well as successful virological control to undetectable setpoint viral loads in a proportion of animals; these results suggest that therapeutic vaccination in combination with TLR7 agonists represents a promising strategy in the search for a functional cure in vivo [104].

Walker-Sperling et al. (2022) assessed the feasibility of a combined approach that included both active and passive immunization in combination with a TLR7 agonist (vesatolimod) to achieving higher rates of virological control following discontinuation of antiretroviral therapy [105]. Only a proportion of *Macaca mulatta* infected with SHIV-SF162P3 that received a combination of a therapeutic vaccination with Ad26 (expressing SIVsmE543 gag-pol, HIV-1 mosaic-1 env, and HIV-1 mosaic-2 env) and MVA (expressing SIVsmE543 gag-pol, HIV-1 mosaic-1 env-gag-pol, and HIV-1 mosaic-2 env-gag-pol) vectors with bNAb PGT101 and a TLR7 agonist rebounded and virological control was achieved in three animals following ART discontinuation [105] (Table 1). Taken together, the studies in non-human primates clearly demonstrated that the use of TLR7 agonists in combination with bNAb or/and therapeutic vaccines has the potential to target latent reservoirs and achieve virological control after discontinuation of antiretroviral therapy.

A novel approach to the activation of TLR-mediated biological effects in the context of HIV vaccine development has recently been reported by Cheng et al. (2022) in an optimisation trial that was based on the fusion of a dendritic cell (DC)-targeting molecule in the DNA vaccine construct in BALB/c and TLR4 knockout C57BL/6 mice [106]. The study used Δ42PD1, a novel alternatively spliced PD-1 (programmed cell death protein-1) isoform that does not activate the PD-L1/2 pathway. Instead, Δ42PD1 activates TLR4 on DCs, inducing their activation and synthesis of proinflammatory cytokines. Since the inadequate immunogenicity represents the most important challenge for the implementation of DNA vaccines, the enhanced antigen immunogenicity and protective efficacy in vivo shown in this study with Δ42PD1-containing vaccines that directly activate TLR4-mediated signalling, represent a promising strategy in the development of DNA vaccines in general [106]

**Table 1 pathogens-12-00092-t001:** Animal studies investigating the role of TLR agonists as latency reversing agents.

Study	Experimental Model	Target TLR	Study Design	Main Outcomes
TLR agonists in humanised mice				
Cheng et al., 2018 [106]	Humanised mice (hu-mice)	TLR3	- therapeutic vaccination of HIV-infected humanised mice with αCD40.HIV5pep vaccine (three highly conserved T-cell epitope regions of HIV Gag, Nef, and Pol fused to the C-terminus of a recombinant anti-human CD40 antibody) in combination with TLR3 agonist poly(I:C)	- poly(I:C) reactivated HIV-1 reservoirs in infected hu-mice - αCD40.HIV5pep with poly(I:C) vaccination induced HIV-specific T cell responses, reduced the concentration of proviral HIV-1 DNA in lymphoid tissues of hu-mice, and significantly delayed time to HIV-1 rebound after ART interruption
TLR agonists in non-human primates				
Lim et al., 2018 [100]	*Macaca mulatta*	TLR7	- administration of TLR7 agonists (GS-986 and GS-9620) in 21 Indian-origin rhesus macaques that were infected with SIVmac251, treated with ART (since 65 days post infection), and experienced complete virological suppression (SIV RNA < 50 copies/mL) for about 400 days (two studies)	- TLR7 agonists induced viral replication in treated animals (up to 1000 SIV RNA copies/mL) and reduced the proviral SIV DNA reservoir (by an average of 75%) - the study failed to demonstrate a difference in time to viral rebound following ART interruption - sustained viral remission longer than 2 years in the absence of ART was observed in two animals following cessation of antiretroviral therapy
Del Prete et al., 2019 [101]	*Macaca mulatta*	TLR7	- administration of TLR7 agonist (GS-9620) in 6 Indian-origin rhesus macaques infected with SIVmac239X, treated with ART at the early stage of infection (since day 13 post infection), and experiencing virological suppression for 75 weeks	- TLR7 agonist failed to induce a measurable increase in plasma viremia, viral RNA-to-viral DNA ratio, or decrease in viral DNA in PBMCs or tissues - SIV-specific CD8+ T-cell responses were not boosted by the TLR7 agonist
TLR agonists and broadly neutralizing antibodies (bNAb) in non-human primates				
Borducchi et al., 2018 [102]	*Macaca mulatta*	TLR7	- a combination of TLR7 agonist (GS-9620) and V3-glycan-dependent bNAb PGT121 in 44 Indian-origin rhesus macaques infected with SIV-SF162P3 (ART initiated during acute infection)	- TLR7 agonist and bNAb combination significantly delayed viral rebound after ART discontinuation - adoptive transfer studies suggested successful targeting of latent viral reservoirs
Hsu et al., 2021 [75]	*Macaca mulatta*	TLR7	- a combination of TLR7 agonist (GS-986) and two bNAbs (N6-LS and PGT121) in 16 Indian-origin rhesus macaques infected with SHIV-1157ipd3N4 (ART initiated at day 14 after inoculation)	- TLR agonist and bNAbs induced SHIV-specific T-cell immunity and immune activation - following ART interruption, median time to viral rebound was significantly shorter in treated vs. untreated animals (6 vs. 3 weeks, respectively)
Mold et al., 2022 [103]	*Macaca mulatta*	TLR7	- a combination of TLR7 agonists (GS-986 and/or GS-9620) and bNAb PGT121 (either as a human IgG1, an effector enhanced IgG1, or an anti-CD3 bispecific antibody) in 33 Indian-origin rhesus macaques infected with SHIV-SF162P3 (ART initiated after 12 months of infection)	- a combination of a TLR7 agonist and all three bNAb was able to prevent viral rebound after treatment discontinuation in a subset of animals
Therapeutic vaccination in combination with TLR agonists in non-human primates				
Borducchi et al., 2016 [104]	*Macaca mulatta*	TLR7	- therapeutic vaccination with an Ad26 vector expressing SIVsmE546 Gag/Pol/Env as prime vaccination and an MVA-based vaccine as a booster in combination with the TLR7 agonist GS-986 - study included 36 Indian-origin rhesus macaques infected with SIVmac251	- significant reduction of median setpoint plasma SIV RNA levels and 2.5-fold delay in time to viral rebound following treatment discontinuation at week 72 - successful virological control to undetectable setpoint viral loads in 3/9 animals
Walker-Sperling et al., 2022 [105]	*Macaca mulatta*	TLR7	- therapeutic vaccination with Ad26-based (expressing SIVsmE543 gag-pol, HIV-1 mosaic-1 env, and HIV-1 mosaic-2 env) and MVA-based (expressing SIVsmE543 gag-pol, HIV-1 mosaic-1 env-gag-pol, and HIV-1 mosaic-2 env-gag-pol) vaccines with bNAb PGT101 and TLR7 agonist GS-9620 - study included Indian-origin rhesus macaques infected with SHIV-SF162P3 (ART initiation at day 9 following infection)	- only 6/10 animals receiving combined active and passive immunisation (vaccine, bNAb, and TLR7 agonist) rebounded and three animals achieved virological control following discontinuation of ART

-antiretroviral therapy (ART), adenovirus serotype 26 (Ad26), modified vaccinia Ankara (MVA); -TLR7 agonist GS-9620 or vesatolimod.

**Table 2 pathogens-12-00092-t002:** Toll-like receptor agonists and HIV: announced clinical trials in humans.

ClinicalTrials.gov Identifier	Study Coordinators	Clinical Trial	Trial Phase	Status (as of 11 November 2022)
TLR agonists and broadly neutralizing monoclonal antibodies				
NCT05281510	Gilead Sciences	A Phase 2a Study to Evaluate the Safety and Tolerability of a Regimen of Dual Anti-HIV Envelope Antibodies, VRC07-523LS and CAP256V2LS, in a Sequential Regimen With a TLR Agonist, Vesatolimod, in Early Antiretroviral-Treated HIV-1 Clade C-Infected Women	Phase 2a	Recruiting
NCT03837756	University of Aarhus	Combining a TLR Agonist With Broadly Neutralizing Antibodies for Reservoir Reduction and Immunological Control of HIV Infection: An Investigator-initiated Randomised, Placebo-controlled, Phase IIa Trial.	Phase 2a	Active, not yet recruiting
Therapeutic vaccination and TLR agonist				
NCT04364035	Aelix Therapeutics	Phase IIa Randomised, Double-blind, Placebo-controlled Study of HIV-1 Vaccines MVA.HTI and ChAdOx1.HTI With TLR7 Agonist Vesatolimod (GS-9620) in Early Treated HIV-1 Infection	Phase 2a	Active, not yet recruiting
NCT04177355	National Institute of Allergy and Infectious Diseases (NIAID)	A Phase 1 Clinical Trial to Evaluate the Safety and Immunogenicity of HIV-1 BG505 SOSIP.664 gp140 With TLR Agonist and/or Alum Adjuvants in Healthy, HIV-uninfected Adults	Phase 1	recruiting
NCT04301154	Henry M. Jackson Foundation for the Advancement of Military Medicine	Phase I, Proof of Concept, Open-Label, Randomized Clinical Trial to Evaluate the Safety and Effects of Using Prime-boost HIVIS DNA and MVA-CMDR Vaccine Regimens With or Without Toll-like Receptor 4 Agonist on HIV Reservoirs in Perinatally HIV Infected Children and Youth	Phase 1	Recruiting
Active and passive immunisation with TLR agonist				
NCT04357821	University of California, San Francisco	Combinatorial Therapy With a Therapeutic Conserved Element DNA Vaccine, MVA Vaccine Boost, TLR9 Agonist and Broadly Neutralizing Antibodies: a Proof-of-concept Study Aimed at Inducing an HIV Remission	Phase 1/phase 2	Active, not yet recruiting

## 5. Current Clinical Trials and Future Research

Following the exceptionally promising results from the studies on TLR7 agonists in non-human primates, the usefulness of this compound has been evaluated in human clinical trials aimed at evaluating its effect on immune system activation as well as the potential to contribute to achieving long-term virological control in HIV-infected patients. SenGupta et al. (2021) recently reported results from a randomised, double-blind, placebo-controlled phase 1b clinical trial on an TLR7 agonist (vesatolimod) in successfully treated HIV-infected persons [107].

An oral TLR7 agonist induced immune activation of DCs and NK cells leading to a decreased quantity of intact proviral DNA in the patients. Lower quantities of intact proviral DNA 13 days after the final dose of the TLR7 agonist also predicted a delayed viral rebound in the patients. These results showed that TLR7 agonists might represent an option in treatment regimens aimed at long-term control of HIV-1. The most recent study by Riddler et al. (2022) reported results on the use of a TLR7 agonist in a double-blind, multicentre, placebo-controlled clinical trial that included 36 successfully treated patients with undetectable plasma viremia [108]. These results showed that TLR7 agonists represent a promising candidate drug aimed at achieving immune stimulation in future clinical trials assessing the efficacy of HIV candidate vaccines. The future of TLR agonists in clinical trials involving therapeutic vaccination and in research into the development of a functional cure for HIV-1 infection looks promising with several studies that are currently recruiting patients or have been announced (Table 2, https://clinicaltrials.gov/).

Currently, there are two clinical trials that will evaluate the role of TLR7 and TLR9 agonists combined with bNAb as potential tools to achieve long-term virological control in HIV-infected patients. An interventional Phase 2a clinical trial (NCT05281510, Gilead) will evaluate a sequential regimen of HIV env-specific monoclonal antibodies VRC07-523LS and CAP256V2LS in combination with the TLR7 agonist vesatolimod (Table 2). The trial will include 25 women infected with HIV subtype C that started antiretroviral therapy at the acute stage of infection. The trial will monitor time to virological rebound, viremia at the end of analytical treatment interruption (ATI), as well as the time to antiretroviral resumption and is estimated to be completed by February 2024. This trial is specifically designed to address the issue of HIV-1 subtype C infection in women. The focus on women is particularly important due to a well-established fact that gender inequality and harmful gender norms restrict women’s access to healthcare services (related and non-related to HIV) and results in a disproportional risk for acquiring HIV infection. According to the UNAIDS epidemiological estimates, there were 260,000 (150,000–390,000) new HIV infections globally among adolescent girls and young women with 83% of these infections occurring in sub-Saharan Africa where adolescent girls and young women (aged 15–24 years) account for 25% of new infections despite representing only 10% of the population. Therefore, clinical trials addressing specific vulnerable groups such as women infected with subtype C are of outmost importance in HIV medicine. The University of Aarhus has announced a phase 2a investigator-initiated, randomised, placebo-controlled clinical trial evaluating the effect of the TLR9 agonist lefitolimod in combination with broadly neutralizing, env-specific, anti-HIV antibodies 3BNC117/10-1074 in HIV-1-infected individuals on ART and during ATI as an intervention to reduce the HIV-1 reservoir. The study is designed to enrol 47 HIV-infected adults (aged 18–65 years) that have been treated with antiretroviral drugs for a minimum of 18 months and had undetectable viremia as well as > 400 CD4 T cells at enrolment. The study will evaluate the sensitivity of the viral reservoir to neutralization using a standardised PhenoSense HIV mAb assay or HIV-1 DNA envelope sequencing in combination with an appropriate bioinformatic algorithm (Table 2).

Two clinical trials will evaluate the effect of TLR7, TLR7/8, and TLR4 agonists as immune adjuvants in therapeutic vaccination protocols. The TLR7 agonist vesatolimod, in combination with a heterologous prime/boost vaccination with HIVACAT T-cell immunogen (HTI) inserted into two vectors (ChAdOx1 and MVA), will be evaluated in a phase 2a, randomised, double-blind, placebo-controlled clinical trial (NCT04364035, Aelix Therapeutics) that has been announced (but not yet recruiting). The study will enrol 57 HIV-infected adults that have started antiretroviral therapy at the early stage of infection with undetectable viremia (for a year) and ≥450 CD4+ T cells/mL (for 6 months) that will undergo treatment interruption. The study will determine the proportion of participants with fully suppressed viremia at 12 and 24 weeks after the start of ATI as well as the number of participants that will achieve virological suppression after resumption of antiretroviral treatment (Table 2). Additionally, a NIAID-led (NCT04177355) phase 1 clinical trial will evaluate the safety and immunogenicity of the HIV-1 BG505 SOSIP.664 gp14 candidate vaccine in combination with the TLR7/8 agonist (3M-052) with or without alum adjuvant in HIV-uninfected healthy adults aged 18–50 years (Table 2). The study will evaluate a number of immunological parameters including serological response to the candidate vaccine with or without a TLR agonist, env-specific CD4+ T-cell immunity, cytokine and chemokine systemic responses, as well as transcriptional profiles of individual cell populations by RNA sequencing analysis. Additionally, in the phase I, open-label, interventional, proof-of-concept, randomised clinical trial NCT04301154 led by the Henry M. Jackson Foundation for the Advancement of Military Medicine, the safety and the effects of combining a TLR4 agonist (as a part of the licenced HPV vaccine Cervarix) with HIVIS DNA plasmids that include *env*, *gag* and *RT*mut/*rev* HIV-1 genes and MVA-CMDR vaccines will be evaluated in perinatally HIV-infected children (Table 2). The scientific background of this trial is the possibility that TLR4, in combination with HIVIS DNA, could increase DNA antigen loading on dendritic cells and promote adaptive immune responses to the HIV vaccine. The effect of TLR4 on vaccine efficacy will be evaluated by monitoring HIV-1 RNA and DNA levels and analysing cellular immunity as well as humoral immunity. The study will include 25 perinatally infected children ≥ 9 years old that have been treated with antiretroviral drugs for at least 6 months with <200 copies of HIV-1 RNA/mL of plasma and is expected to continue until October 2023.

A combination of active and passive immunisation and TLR agonists that has shown potential in non-human primate studies and one clinical trial in humans will evaluate this concept in the future. A phase 1/phase 2, proof-of-concept, single arm study (NCT04357821, University of California, San Francisco) that will be conducted on 20 successfully treated HIV-infected adult patients receiving several regimens containing IL-12 adjuvanted p24CE DNA prime, IL-12 adjuvanted DNA boost (p24CE plus p55gag), MVA/hiv62Bboost, two binding monoclonal antibodies VRC07-523LS and 10-1074 that target the CD4 binding site and V3 loop, respectively, and the TLR9 agonist lefitolimod following ATI with a single dose of both binding antibodies (Table 2). It is expected that the patients will experience a viral rebound a few weeks after the concentration of binding monoclonal antibodies is reduced to the sub-therapeutic level but hopefully it will be associated with a lower viral load set-point and long-term remission.

## 6. Conclusions

Basic knowledge on the role of TLR in innate immunity against HIV has been translated into in vivo animal studies and clinical research in humans. Agonists of TLR7, TLR7/8, and TLR4 in combination with bNAb, therapeutic vaccines, or both represent an exceptionally promising strategy for achieving long-term remission without antiretroviral treatment or a functional cure for HIV-infected persons. Despite the large body of evidence, the role of TLR SNPs in the immune response to HIV and its potential clinical significance is still not fully understood due to the complex interactions between the observed genetic variants, environmental factors, ethnicity, gender, and the possibility of multigenic effects. The most promising results in the use of TLR agonists as LRAs have been achieved with TLR7 agonists in combination with bNAb. Additionally, the use of TLR7 agonists has shown positive results in clinical trials involving active and passive HIV-1 vaccination.

## Figures and Tables

**Figure 1 pathogens-12-00092-f001:**
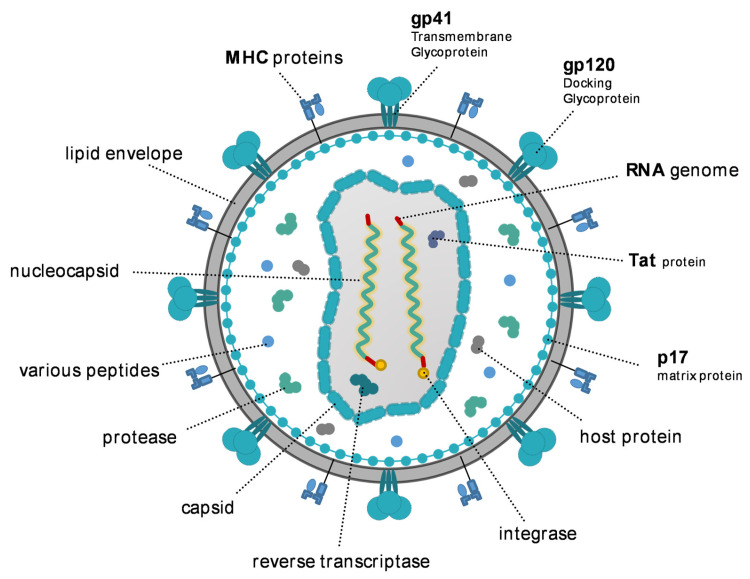
Structure of HIV-1.

**Figure 2 pathogens-12-00092-f002:**
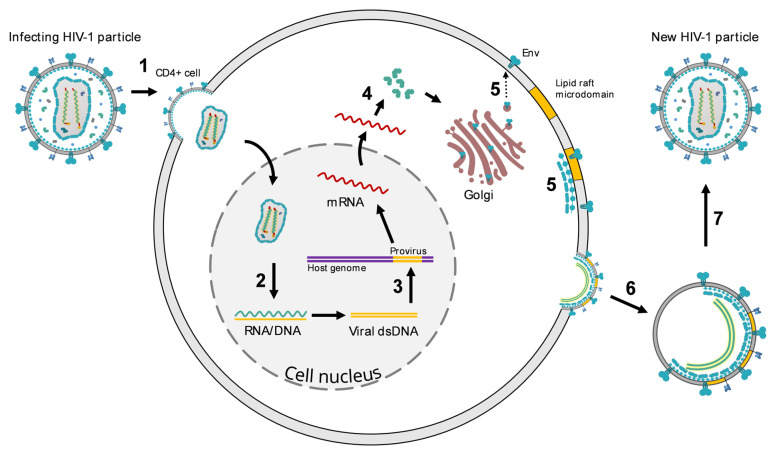
Mechanism of HIV-1 infection. (**1**) Binding, fusion, and entry of the HIV-1 virion into the host cell. (**2**) Uncoating of viral single-stranded (ssRNA) and reverse transcription into DNA. (**3**) Integration of synthesised double-stranded (dsDNA) into the host genome (provirus). (**4**) Protein synthesis and assembly. (**5**) Virion assembly. (**6**) Release of new HIV-1 virion. (**7**) Maturation.

**Figure 3 pathogens-12-00092-f003:**
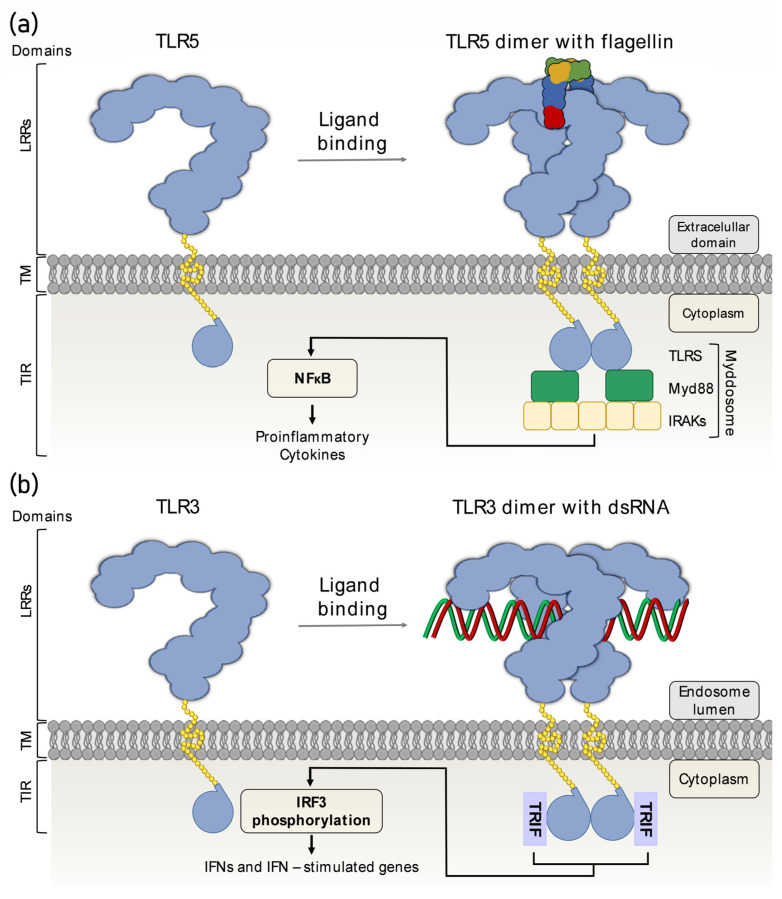
Induction of signalling TLRs. (**a**) TLR5 is expressed on the cell membrane; after binding its ligand flagellin, dimerization of the two TLR5 receptors occurs. Dimerization of the TIR-domains causes the recruitment of Myd88 and, along with IRAKs, formation of the myddosome. This initiates a signalling pathway that activates NF-κB and leads to the synthesis of proinflammatory cytokines. Other TLRs expressed on the cell membrane including TLR5, TLR1, TLR2, TLR4, TLR6, and TLR10 initiate the Myd88 signalling pathway. (**b**) TLR3 is the only receptor that strictly uses the TRIF signalling pathway. TLR3 is positioned on the endosomal membrane and is activated by binding of its ligand, dsRNA. Followed by ligand binding, dimerization of TLR3 occurs. TLR3 uses the TRIF signalling pathway where TRIF recruits IKKε and TBK1, leading to activation of NF-κB, IRAK3, and AP-1 and, consequently, synthesis of proinflammatory cytokines and type I interferons (type I IFNs). Other TLRs on the endosomal membrane are TLR7, TLR8, and TLR9. However, they use the Myd88 signalling pathway while TLR3 is the only receptor that strictly uses the TRIF-signalling pathway. TLR4 can use both the Myd88 signalling pathway and TRIF signalling pathway.

## Data Availability

Not applicable.

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
