# Peer review of "Role of TLRs in HIV-1 Infection and Potential of TLR Agonists in HIV-1 Vaccine Development and Treatment Strategies"

_pathogens, 2023, doi:10.3390/pathogens12010092_

Round 1

Reviewer 1 Report

In this manuscript, the authors summarized the knowledge of TLRs as well as the development of TLR agonists as potential HIV therapeutic agents. Overall, this is a well-organized and well-written review which comprises sufficient background knowledge and updated discoveries, and it could be processed for the publication in Pathogens. I listed the following comments for the authors’ consideration and hopefully they could help to improve the quality of the manuscript.

Major points:

1 In Figure 2, the step 4 only showed the translation of HIV proteins and their assembly. The newly transcribed HIV genome assembled into virions should also be depicted.

2 The polymorphisms and SNPs of TLRs (section 3.4) contained a large amount information, which would be more straightforward to be digested with a table. In addition, there was no “Table 1” in manuscript, why the table in page 18 was labeled “Table 2”?

3 The citation in line 418 (ref 62) was incorrect, which should be ref 45. Why there were some non-English characters in references, such as ref 65 in line 987 and ref 83 in line 1027? In addition, the format of some references was wrong and incomplete, such as ref 95 in line 1050, the title of this paper was missing. The format of whole references should be revised carefully.

4 The text in line 475-489 was not associated with TLR10 (4p14), so this part could be separated as a new section.

Minor points:

1 In line 53, “…toll-like receptors (TLRs) have been recognized as potential LRAs…” should be “…toll-like receptors (TLRs) agonists…”.

2 The sentence in lines 58-60 was too long and confusing.

3 The format of genes names should be italicized, such as HIV genes in page 3.

4 Some formats in manuscript were inconsistent. For example, the title in line 148 had a right bracket after the number, while in other parts such as line 67, it was a dot.

5 The font size of “TRIF” and “IRF3 phosphorylation” in Fig 3b was too small.

6 There was a typo in line 507. “TL2” should be “TLR2”.

7 In line 572, “TLRs” should be “TLR agonists”.

8 Grammatical errors were occasionally seen.

Author Response

Reviewer 1:

In this manuscript, the authors summarized the knowledge of TLRs as well as the development of TLR agonists as potential HIV therapeutic agents. Overall, this is a well-organized and well-written review which comprises sufficient background knowledge and updated discoveries, and it could be processed for the publication in Pathogens. I listed the following comments for the authors’ consideration and hopefully they could help to improve the quality of the manuscript.

Major points:

1 In Figure 2, the step 4 only showed the translation of HIV proteins and their assembly. The newly transcribed HIV genome assembled into virions should also be depicted.

Author's response: Figure 1. is modified, virion assembly is now shown following HIV-1 proteins translation.

2 The polymorphisms and SNPs of TLRs (section 3.4) contained a large amount information, which would be more straightforward to be digested with a table. In addition, there was no “Table 1” in manuscript, why the table in page 18 was labeled “Table 2”?

Author's response: Table 1 is now added as a suplement table.

3 The citation in line 418 (ref 62) was incorrect, which should be ref 45. Why there were some non-English characters in references, such as ref 65 in line 987 and ref 83 in line 1027? In addition, the format of some references was wrong and incomplete, such as ref 95 in line 1050, the title of this paper was missing. The format of whole references should be revised carefully.

Author's response:  The citation in line 418 is changed to the reference 45. We found no non-English characters in the reference 65. Reference in line 1027 had non-English characters because of anomaly that occured in Mendeley program through which references were added, this is now corrected and no non-English characters are present in the reference. Title of the reference 95 is added, format of the rest of the references was revised as well.

4 The text in line 475-489 was not associated with TLR10 (4p14), so this part could be separated as a new section.

Author's response:  Since only first sentence was associated with TLR10, the title of whole section is changed to „Significance of TLR polymorphisms“.

Minor points:

1 In line 53, “…toll-like receptors (TLRs) have been recognized as potential LRAs…” should be “…toll-like receptors (TLRs) agonists…”.

Author's response: The line 53 is corrected.

2 The sentence in lines 58-60 was too long and confusing.

Author's response:  The sentence in lines 58-60 is now separated into 3 sentences which makes content of the sentence more clear.

3 The format of genes names should be italicized, such as HIV genes in page 3.

Author's response: Format of genes is now italicized.

4 Some formats in manuscript were inconsistent. For example, the title in line 148 had a right bracket after the number, while in other parts such as line 67, it was a dot.

Author's response: Format is corrected, dot is placed next to the chapter number through the whole manuscript.

5 The font size of “TRIF” and “IRF3 phosphorylation” in Fig 3b was too small.

Author's response:  Font size is changed to the bigger one.

6 There was a typo in line 507. “TL2” should be “TLR2”.

Author's response:  Typo is corrected.

7 In line 572, “TLRs” should be “TLR agonists”.

Author's response:  „TLRs“ is changed to „TLR agonists“.

8 Grammatical errors were occasionally seen.

Author's response:  Grammatical errors are corrected.

Reviewer 2 Report

This review summarizes the role of TLRs in the innate immunity to HIV-1, TLR gene polymorphisms in HIV, and the therapeutic attempts made for the use of TLR agonists as latency reversing agents. The review appears to be informative. However, several concerns are noted, and comments were made below to increase the quality and clarity of the contents in the manuscript.

1. In the section of TLR agonists as latency reversing agents, some results are not well explained/discussed. For example, it is stated that GEM91 showed contradictory results with reduction of HIV replication but increased viremia (lines 600-602). This ended without any possible explanation and discussion about the results. This section can be expanded as to the role of TLR medicated immune responses and the outcome in the individual studies.

2. It is suggested that the authors change the numbering with the associated content in a more logical way and increase the readability. The section of TLR (#3) can be divided into two categories such as 1) basic role of TLRs in host immunity and 2) TLR functions in HIV separately. The section of #4 (activation of latent HIV-1 reservoir) is not well structured in terms of their subsections. 4.1 and 4.1.1. are mixed up and densely written. Section 4.1 can be briefly summarized with more detailed descriptions in several subsections of 4.1.1., 4.1.2., etc.

3. The manuscript requires proofreading to fix errors. Some of the examples, but not all, are listed below:

a. Please correct all the errors found in the spacing between words.

b. Line 55: change “proteins” to another proper word such as components or elements.

c. Check the section labeling: 1, 2, 3, not 3), 4, not 4), etc. to be consistent.

d. Figure 1: figure legend is missing. Check the spelling – “capside”? What is “peptide”?

e. Figure 3: “TRIF” letter is not well visible in the figure. In the figure legend, line 249, change “starts”, which does not come with objects, to another word such as initiates or triggers.

f. Line 316, “propose the as a risk…”?

g. Line 601, what does designate “both”?

Author Response

Reviewer 2

This review summarizes the role of TLRs in the innate immunity to HIV-1, TLR gene polymorphisms in HIV, and the therapeutic attempts made for the use of TLR agonists as latency reversing agents. The review appears to be informative. However, several concerns are noted, and comments were made below to increase the quality and clarity of the contents in the manuscript.

  1. In the section of TLR agonists as latency reversing agents, some results are not well explained/discussed. For example, it is stated that GEM91 showed contradictory results with reduction of HIV replication but increased viremia (lines 600-602). This ended without any possible explanation and discussion about the results. This section can be expanded as to the role of TLR medicated immune responses and the outcome in the individual studies.

Author's response:  Section about GEM91 is expanded and explained in more detail.

  1. It is suggested that the authors change the numbering with the associated content in a more logical way and increase the readability. The section of TLR (#3) can be divided into two categories such as 1) basic role of TLRs in host immunity and 2) TLR functions in HIV separately. The section of #4 (activation of latent HIV-1 reservoir) is not well structured in terms of their subsections. 4.1 and 4.1.1. are mixed up and densely written. Section 4.1 can be briefly summarized with more detailed descriptions in several subsections of 4.1.1., 4.1.2., etc.

Author's response:   Section 3.5. „Interaction between TLRs and immune system“ was moved ahead as 3.4. section with title changed to „Basic role of TLRs in host immunity“ and expanded to section 3.4.1. „Recognition of nucleic acids by TLRs“. Afterwards comes the section „The role of TLR receptor polymorphisms in the biology and in the clinical aspects of the HIV infection” as new section 3.5. Start of the section  3.”Toll-like receptors “ was changed in the following way: 3.1.Types of TLRs in humans, 3.2.Structure and position, 3.3. Mechanism of action, as suggested by Reviewer 3. Sections 4.1. have been condensed and shortened.

  1. 3. The manuscript requires proofreading to fix errors. Some of the examples, but not all, are listed below:

Author's response:   

  1. Please correct all the errors found in the spacing between words.

Corrected

  1. Line 55: change “proteins” to another proper word such as components or elements.

„Proteins“ is changed to „elements“.

  1. Check the section labeling: 1, 2, 3, not 3), 4, not 4), etc. to be consistent.

Section labeling is now consistent throughout the manuscript.

  1. Figure 1: figure legend is missing. Check the spelling – “capside”? What is “peptide”?

„Capside“ is changed to „capsid“. „Peptide“ was changed to „various peptides“ reffering to the ones pickep up from the host.

  1. Figure 3: “TRIF” letter is not well visible in the figure. In the figure legend, line 249, change “starts”, which does not come with objects, to another word such as initiates or triggers.

„TRIF“ is more visible now. „Starts“ is changed to „initiates“.

  1. Line 316, “propose the as a risk…”?

This sentence is now removed.

  1. Line 601, what does designate “both”?

„Both“ is removed as it was typed by mistake.

Reviewer 3 Report

The review by Rozman M et al focuses on the role of Toll-like receptors in HIV-1 infection with special emphasis on TLR gene polymorphisms and the application of TLR agonists in HIV-1 vaccine development. It’s a well-composed review covering several aspects of TLR function in the immune response against HIV and could be a useful contribution to the field. The content fits into the readership of Pathogens.

I would suggest addressing the following concerns before accepting the manuscript. 

1)    Authors should consider editing the English language and style all throughout the review. There are several lengthy and complex sentences that can be confusing sometimes, I would suggest making small but clear sentences for clarity. Many sentences need rephrasing; some of these are highlighted here, lines 105-108, 130-133, 177-179, 280-282, 415-418, 439-441, 460-463, 472-473, 487-489, 534-537, 566-569, 577-582, 593-596, 624-627, 655-658

2)    Section 2 is quite big and over-descriptive and somewhat deviates from the main aim of the review. Authors should consider making it more concise. Lines 121-124 and 124-128 convey a similar meaning. Rephrasing will make it clearer.

3)    Section 3 needs revision. First, beginning with the types of TLRs followed by structure and function and mechanism of action, and so on would be more appropriate. Sections 3.3 and. 3.3.1 could be merged.

4)    Figure 3 depicts only the signaling pathways induced by TLR3 and 5, however, if the mechanism is similar for other TLRs, it’s better to mention their names in the figure itself, like categorizing them into two groups: cell membrane-based TLRs and endosome-based TLRs signaling. 

5)    Section 3.4 title could be more specific. What’s the focus of this section, the role of TLR polymorphisms on different diseases biology or only on HIV-1 infection? Depending on the purpose of the section, authors are advised to edit the title and the content.

6)    Section 4.1 title should be ‘TLR agonists as latency-reversing agents.’ 

7)    Incorporating tables for sections 3.4 and 4.1 rather than so much text will enhance the review’s impact and presentation.  

8)    References missing from several lines.  

9)    Abbreviations should be used all throughout the review once the expanded full form has been mentioned at the beginning. 

10) Grammar needs to be checked thoroughly as there are several mistakes. Lines 349, 365-366, 428, 488-489, 566, 625-626, 693, 697, 713, 718, etc.

Author Response

Reviewer 3:

The review by Rozman M et al focuses on the role of Toll-like receptors in HIV-1 infection with special emphasis on TLR gene polymorphisms and the application of TLR agonists in HIV-1 vaccine development. It’s a well-composed review covering several aspects of TLR function in the immune response against HIV and could be a useful contribution to the field. The content fits into the readership of Pathogens.

I would suggest addressing the following concerns before accepting the manuscript. 

1)    Authors should consider editing the English language and style all throughout the review. There are several lengthy and complex sentences that can be confusing sometimes, I would suggest making small but clear sentences for clarity. Many sentences need rephrasing; some of these are highlighted here, lines 105-108, 130-133, 177-179, 280-282, 415-418, 439-441, 460-463, 472-473, 487-489, 534-537, 566-569, 577-582, 593-596, 624-627, 655-658

Author's response: The English language and style have been changes throughout the review. Due to the extend of the changes, please see the corrected version of the manuscript with track changes.

2)    Section 2 is quite big and over-descriptive and somewhat deviates from the main aim of the review. Authors should consider making it more concise. Lines 121-124 and 124-128 convey a similar meaning. Rephrasing will make it clearer.

Author's response: Section 2 (as well as section 1) were re-written and shortened to make the review a bit more focused.

3)    Section 3 needs revision. First, beginning with the types of TLRs followed by structure and function and mechanism of action, and so on would be more appropriate. Sections 3.3 and. 3.3.1 could be merged.

Author's response: The order in section 3 is arranged in the suggested way: 3.1.“Types of TLRs in humans“, 3.2.“Structure and position“, 3.3. „Mechanism of action“. Sections 3.3. and 3.3.1. were changed due to suggestion by Reviewer 2 in a way that Section 3.5. „Interaction between TLRs and immune system“ was moved ahead as 3.4. section with title changed to „Basic role of TLRs in host immunity“ and expanded to section 3.4.1. „Recognition of nucleic acids by TLRs“. Afterwards comes the section „The role of TLR receptor polymorphisms in the biology and in the clinical aspects of the HIV infection” as new section 3.5.

4)    Figure 3 depicts only the signaling pathways induced by TLR3 and 5, however, if the mechanism is similar for other TLRs, it’s better to mention their names in the figure itself, like categorizing them into two groups: cell membrane-based TLRs and endosome-based TLRs signaling. 

Author's response: TLRs are, according to their position, divided into TLRs on cell-membrane and the TLRs on endosomal membrane. However, position of TLR doesn't define its signaling pathway with TLR3 being the only receptor that uses strictly TRIF-mediated one. All other TLRs use Myd88-signaling pathway regardless of position with TLR4 being able to initiate both. Therefore, suggested categorization was not possible.  However, more detailed explanation was given in the figure legend. Following description of each signalling pathway, all TLRs using described pathway are now listed as well as their position.

5)    Section 3.4 title could be more specific. What’s the focus of this section, the role of TLR polymorphisms on different diseases biology or only on HIV-1 infection? Depending on the purpose of the section, authors are advised to edit the title and the content.  

Author's response: Significant changes have been made to this part of the text and the studies focusing on other infectious diseases have been removed (only studies on HIV and important co-infections have been kept).

6)    Section 4.1 title should be ‘TLR agonists as latency-reversing agents.’ 

Author's response: „TLRs“ was changed to „TLR agonists“.

7)    Incorporating tables for sections 3.4 and 4.1 rather than so much text will enhance the review’s impact and presentation.

Author's response: Based on your suggestion, tables for both sections have been added. The text has been significantly edited and made shorter.   

8)    References missing from several lines. 

Author's response: corrected

9)    Abbreviations should be used all throughout the review once the expanded full form has been mentioned at the beginning. 

Author's response: corrected

10) Grammar needs to be checked thoroughly as there are several mistakes. Lines 349, 365-366, 428, 488-489, 566, 625-626, 693, 697, 713, 718, etc.

Author's response: corrected

Round 2

Reviewer 1 Report

I sincerely appreciate the authors' responses to my comments and revision of the whole manuscript. My former concerns have been clarified, and in my opinion the manuscript now can be considered for acceptance.

Author Response

Dear reviewer, thank you very much for your positive response to our revised version of the manuscript. With kind regards (authors)

Reviewer 3 Report

Dear Authors,

There are still some minor changes that need to be done. Please find my comments below.

Line 59, change ‘To date, only several’ to ‘To date, only some.’

Lines 65 and 69, begin with ‘Therefore.’ Avoid using the same adverb in consecutive sentences.

Lines 69-72, 391-392, and 518-519 are unclear. Modify it.

Typo error in lines 390, 733, 863, 909, and 923.

Line 498- ‘while the association ‘with’ HIV infection.

Line 744- The abbreviation for peripheral blood mononuclear cells is PBMCs.

References missing from several lines (for example 779-781, 837-840, 969, 982, 993, 1010), mostly at the end of sentences.

Line 800-802- ‘Previous phase trials results showed its success in both latency reversal and immunity induction.’

Line 811- ‘The concept of the vaccine was based on targeting…’

Still, in several lines, abbreviations and expanded forms are used together ( for example, lines 821 and 825) Kindly check it thoroughly.

Line 841- No need to change the paragraph. Line 841, 991.

Line 992- Include more information about HIVIS DNA?

Quote reference numbers in tables.

Information about the animal model and TLR agonist missing from Table 2, Walker- Sperling et al, 2022. 

Author Response

Dear Authors,

There are still some minor changes that need to be done. Please find my comments below:

- Line 59, change ‘To date, only several’ to ‘To date, only some.’

Author’s response: Corrected

- Lines 65 and 69, begin with ‘Therefore.’ Avoid using the same adverb in consecutive sentences.

Author’s response: Corrected

- Lines 69-72, 391-392, and 518-519 are unclear. Modify it.

Author’s response: The sentences in lines 69-72 and 391-392 have been modified and the sentence in lines 518-519 has been deleted.

- Typo error in lines 390, 733, 863, 909, and 923.

Author’s response: Corrected

- Line 498- ‘while the association ‘with’ HIV infection.

Author’s response: Corrected

- Line 744- The abbreviation for peripheral blood mononuclear cells is PBMCs.

Author’s response: Corrected

- References missing from several lines (for example 779-781, 837-840, 969, 982, 993, 1010), mostly at the end of sentences.

Author’s response:

The reference for all clinical trials that have been either announced or are already recruiting patients (described in lines 969, 928, 993 and 1010) is www. https://clinicaltrials.gov/ (provided at the end of the sentence in line 939 that preceeds the text describing them). 

- Line 800-802- ‘Previous phase trials results showed its success in both latency reversal and immunity induction.’

Author’s response: Corrected

- Line 811- ‘The concept of the vaccine was based on targeting…’

Author’s response: Corrected

- Still, in several lines, abbreviations and expanded forms are used together ( for example, lines 821 and 825) Kindly check it thoroughly.

Author’s response: Corrected

- Line 841- No need to change the paragraph. Line 841, 991

Author’s response: Corrected for line 841 (line 991 was not changed)

- Line 992- Include more information about HIVIS DNA?

Author’s response: The information about the genetic composition of HIVIS DNA has been included in the text.

- Quote reference numbers in tables.

Author’s response: Reference numbers have been included in the tables (except in the one describing the studies announced clinical trials- the link reffering to them has been included in the manuscript)

- Information about the animal model and TLR agonist missing from Table 2, Walker- Sperling et al, 2022. 

Author’s response: The information has been included.